# Computed Tomography Evaluation of Normal Canine Abdominal Lymph Nodes: Retrospective Study of Size and Morphology According to Body Weight and Age in 45 Dogs

**DOI:** 10.3390/vetsci8030044

**Published:** 2021-03-07

**Authors:** Simone Teodori, Giovanni Aste, Roberto Tamburro, Antonio Maria Morselli-Labate, Francesco Simeoni, Massimo Vignoli

**Affiliations:** 1Roma Sud Veterinary Clinic, via Pilade Mazza 24, 00173 Rome, Italy; si.teodori@gmail.com; 2Faculty of Veterinary Medicine, University of Teramo, Piano d’Accio, 64100 Teramo, Italy; fsimeoni@unite.it (F.S.); mvignoli@unite.it (M.V.); 3Biostatistic, via Battibecco 1, 40123 Bologna, Italy; antoniomaria.morsellilabate@gmail.com

**Keywords:** computed tomography, lymph nodes, dog

## Abstract

The morphological characteristics of the largest lymphatic vessels and lymph nodes of the body have been described through ultrasonography, although food and gas in the gastrointestinal tract can often have negative effects on the response of small abdominal structures. The aim of the study was to describe the size of normal abdominal lymph nodes (ALs) in dogs affected by disease, not including lymphadenomegaly or lymphadenopathy, and divided according to body weight and age. The ALs studied included the jejunal, medial iliac, portal, gastric, splenic, and pancreaticoduodenal lymph nodes. Statistical correlation considering body weight and age as continuous variables showed that all measurements of the ALs increased according to body weight changes (*p* < 0.01). The most reliable values were the volume measurements (*p* < 0.001) compared to the length, thickness, and width. Mixed results emerged from a comparison of weight categories and age; only the jejunal lymph nodes showed a significant correlation (*p* < 0.05). Other characteristics (shape, attenuation, and enhancement) are subsequently reported. The resulting data can be used to categorize CT measurements of normal ALs displayed based on the body weight and age of the subjects. This study aimed to propose a new parameter of normalcy that may serve as a reference for the evaluation of infectious or neoplastic events.

## 1. Introduction

The largest lymphatic vessels and lymph nodes of the body are described regionally according to the following categories: head and neck, thoracic limb, thorax, abdominal and pelvic walls, genital organs, abdominal viscera, and pelvic limb. The abdomen and pelvis, such as the chest, can be divided into a parietal group and a visceral group. The parietal group includes the lymph-node center of the abdominal and pelvic walls: the lumbar, iliosacral, and iliofemoral centers. The visceral group is subdivided into subgroups that apply to specific organs: celiac, cranial, and caudal mesenteric [1]. For many of these lymph nodes, morphological characteristics have been described through ultrasonography, although food and gas in the gastrointestinal tract can often have negative effects on the response of small abdominal structures [2,3,4,5,6,7,8,9,10]. There are studies for tracheobronchial, sternal, or cervical lymph nodes [11,12,13,14,15] that have instead proposed computed tomography (CT) as a modality of investigation, and recently, CT was also proposed for morphological and morphometric evaluation of normal abdominal lymph nodes (ALs) to reduce the limitations of ultrasound [16]. Until recently, characteristics studied by CT of the ALs have been described only in relation to specific abdominal pathologies [11]. Considering limited information in the literature, the aim of this retrospective study was to assess the relationship between age, body weight and lymph node features in dogs with normal lymph nodes. Other secondary objectives were to increase knowledge regarding the characteristics (shape, attenuation, and enhancement) of these structures in CT in order to provide a reference for evaluation of infectious or neoplastic events.

## 2. Materials and Methods

For all the examinations there was the approval of the owners by informed consent signature. All the clinical procedures and the care of the animals complied to the national legislation on animal care (Legislation decree n.26, 03/03/2014) and adhered to the internal rules of University of Teramo.

### 2.1. Animals

Data from all subjects who underwent CT total body in direct scanning and after contrast administration at the Policlinico Veterinario Roma Sud between December 2014 and December 2017 were analyzed.

### 2.2. Inclusion Criteria

Dogs were selected for analysis according to the following inclusion criteria: achievement of the eighteenth month of life; absence of alterations compatible with an inflammatory or neoplastic process involving the abdomen, pelvis, lower limbs, abdominal, or perineum wall; absence of malignant or multicentric processes on the remaining parts of the body with the possibility that metastases could be found in the abdominal organs, pelvis, lower limbs, abdominal or perineum wall; absence of pleural and/or abdominal effusion; absence of movements and/or breathing artefacts; a BCS (body condition score) between 2/5 and 4/5; regarding traumatized subjects, only if less than 12 h had elapsed since the traumatic event; absence of alterations to blood and urine tests; and absence of anti-inflammatory therapies in the previous 10 days. Standard CVRS laboratory values were used as a reference for CBC and for the following blood chemistry examination parameters: albumin, ALKP, ALT, AST, CPK, GGT, amylase, lipase, BUN, creatinine, phosphorus, calcium, cholesterol, glucose, bilirubin, total proteins, and globulins.

### 2.3. Experimental Design

Dogs were divided into three categories (including 15 specimens each) according to a previous review [16] for some dimensional evaluations of the small abdominal structure compared to the dog weight. Thereby, the “S” group consisted of dogs weighing less than or equal to 10 kg; the “M” group consisted of animals with a weight between 10 and 30 kg; and the “L” group consisted of dogs with a weight greater than or equal to 30 kg. As far as age was concerned, dogs were dichotomized into “youths” (dogs aged between 18 and 24 months) and “adults” (dogs aged more than two years).

### 2.4. Procedures

The CT procedures were performed under general anesthesia obtained through Fentanyl (0.1 mg/kg IV) as premedication, propofol (5–6.5 mg/kg IV, Propofol Kabi 20 mg/mL, Fresenius Kabi Italia Srl. Verona, Italy) for induction and isoflurane mixed with oxygen (Isoflo 250 mL 100% p/p, Zoetis Italia Srl. Queensborough, UK) administered via gaseous endotracheal tube for maintenance. Transient apnea was then induced by approximately 60 s of hyperventilation and a bolus of fentanyl before each scan. For all subjects, tomographic images were obtained through the use of a 16-slice CT (Philips MX 16 Slice, Philips Medical Systems, Best, The Netherlands) and through the use of the standard logarithm for the acquisition of both direct scans and post-administration of the contrast medium (venous phase). Contrast medium (600 mg/kg IV, OptirayTM 300 mg/mL Injectable solution, Ioversol, Intravasal use, Pre-filled syringe. 125 mL. Mallinckrodt Pharmaceuticals Italia Srl, Milan, Italy) was injected via an injector (Liebel-Flarsheim CT 9000 ADV Injector CT. Liebel-Flarsheim Company LLC, Cincinnati, OH, USA) at a rate of 3 mL/s through 20–22 gauge intravenous catheters on the right or left cephalic vein. Dogs lay the sternal decubitus with the anterior and posterior limbs extended; the scan parameters were 120 kV, 190 mA, 1–2 mm slice thickness, pitch of 1:1 and 0.6 s/rotation. Measurement values were calculated using a soft tissue window with a width of 350 (WW, window width) and a level of 40 (WL, window level).

### 2.5. Measurements

The ALs analyzed were two jejunal, one hepatic, one splenic, one gastric, one pancreaticoduodenal, and two medial iliacs. For each lymph node, the localization was recorded by a single observer (ST) based on the anatomical indications present in the literature [6] and, subsequently, a second observer (MV) oversaw and gave consent for data obtained. Dimensions (length, thickness, and width), the volume and X-ray attenuations were evaluated.

The length was defined as lymph node’s maximum size measured in a cranio-caudal direction. The thickness and width were measured in a dorsoventral direction and laterolateral sense perpendicular to the length, respectively. Software (Horos, Horosproject, Horos v2.0.2, www.horosproject.org accessed on 18 March 2018) was used to estimate the AL total volume through slice selected areas. The area was calculated on transverse images by the perimeter of a selected structure. This calculation was performed on every single consecutive image that included the selected structure. The perimeter was defined manually (Figure 1).

Finally, shapes were recorded by stratifying ALs into three categories: (a) elongated, (b) roundish, and (c) mixed. Shape was judged in a subjective way through 3D volume rendering reconstructions with an evaluation similar to that reported for ultrasonography [17]. Bilobated or multilobed lymph nodes were considered to be elongated. When they could not be clearly entered in one of the two categories, they were categorized as having mixed shape.

Attenuation was measured on the same transverse image, before and after contrast administration, using an oval or round region of interest (ROI) delineated as extensively as possible. Immediately afterwards, if enhancement could distinguish between cortical and medullary areas, the structure was defined as heterogeneous. When contrast was distributed uniformly throughout the entire structure, the enhancement was described as homogeneous.

### 2.6. Statistical Analysis

Median and range (minimum and maximum) values were used to describe scalar variables, while absolute and relative frequencies were reported for discrete variables. Data were stratified ingroups in order to describe the values of body weight (three categories) and age (two categories), while the statistical correlation analysis was achieved by considering continuous data. Non-parametric statistics were used in order to analyze scalar variables: the Kruskal–Wallis test was applied for comparing groups and the Spearman rank correlation test was applied for testing relationships with body weight and age. In particular, Rho correlation was considered as follow: less than 0.3 as weak, from 0.3 to 0.5 as moderate, and greater than 0.5 as strong correlation.

Data were managed and analyzed by using the SPSS Statistics package (version 23 Software for Windows, IBM Co., Armonk, NY, USA): two-tailed *p* values less than 0.05 were considered statistically significant.

## 3. Results

From the archive, 122 dogs were identified based on the CT examination, but only 45 of these dogs were chosen based on the completeness of the medical records, which fully satisfied the inclusion/exclusion criteria during the enrolment period and therefore corresponded to a negative result for abdominal lymphadenopathies.

The CT scan was performed in 34 dogs with acute thoraco-lumbar spine injury (discopathy) or traumatic events (subluxation, luxation, or fracture); in 4 dogs CT was carried out to exclude conditions related to Horner’s syndrome. Finally seven dogs were examined by CT for follow-up six months after thoracic surgery (lobectomy) performed for treatment of previous pneumothorax.

Median body weight was 20 kg (range 2–62 kg). Dogs were divided into three categories, including 15 specimens each, based on body weight. The “S” group (dogs weighing less than, or equal to, 10 kg) showed a median weight of 6.3 kg (range 2–9 kg); the “M” group (animals with a weight between 10 and 30 kg) had a median weight of 20 kg (range 12–29 kg); and finally, the “L” group (dogs with a weight greater than, or equal to, 30 kg) had a median weight of 35 kg, (range 30–62 kg).

Median age of all dogs included was 5 years (range 1.5–13 years), with overlapping results between various weight categories. Twelve (26.7%) dogs were “youths”, and 33 (73.3%) were “adults”. The median age for the “S” group was 9 years (range 2–13 years), for the “M” group it was 5 years (range 1.5–13 years), and for the L group it was 4 years (range 1.5–11 years). No significant differences among the groups were noted (*p* = 0.360).

Out of the 45 dogs selected, 32 (71.1%) were males (6 neutered), and 13 (28.9%) were females (1 neutered). There were 9 males and 6 females in the “S” group, among which 1/15 (6.7%) were sterilized animals; in the “M” group, there were 10 males and 5 females, among which 2/15 (13.3%) were sterilized; in the “L” group, there were 13 males and 2 females, among which 4/15 (26.7%) were sterilized. Additionally, regarding sex, a homogeneous distribution between the weight groups was noted.

The study included mixed-breed dogs (*n* = 10), English Cocker Spaniels (*n* = 3), Labrador Retrievers (*n* = 3), Jack Russell Terriers (*n* = 2), Maltese (*n* = 2), Dachshunds (*n* = 2), Rottweilers (*n* = 2), Weimaraners (*n* = 2), Boxers (*n* = 2), a Poodle (*n* = 1), a West Highland white terrier (*n* = 1), a German Pinscher (*n* = 1), a Chihuahua (*n* = 1), a Cavalier King Charles spaniel (*n* = 1), a Border Collie (*n* = 1), an Irish Setter (*n* = 1), a Beagle (*n* = 1), a Basenji (*n* = 1), a German Shepherd (*n* = 1), a Course Retriever (*n* = 1), a Dalmatian (*n* = 1), an Argentinian Dogo (*n* = 1), a Golden retriever (*n* = 1), a Dobermann (*n* = 1), and a Flat Coated Retriever (*n* = 1).

Four-hundreds and two ALs (out of the total of 405 considered ALs) were visualized and measured, exhibiting positioning abnormalities and variability already described by Beukers et al. [16]; 3 lymph nodes (one hepatic in the L category, one splenic in the L category and one gastric in the S category) were not found. Almost all of the liver lymph nodes were visualized near the hepatic hilum (Figure 2), often symmetrically distributed with one to the right and one to the left of the portal vein. In some cases, two lymph nodes were found along the portal vessel, both on the same side, while in one case, it was not possible to identify more than one.

Splenic lymph nodes were usually found along the dorsal margin of the splenic vein (Figure 3), and almost all of them were disposed with their major axis along the subject’s transverse plane rather than on the sagittal plane, as most ALs have been explored.

All gastric lymph nodes were found medially with respect to the small gastric curvature (Figure 4) in the passage area between the body and pylorus.

A pancreaticoduodenal lymph node was always found in the right pancreatic lobe region (Figure 5).

At least two medial iliac lymph nodes were visualized and measured in each subject, one on the right and one on the left (Figure 6).

These were located at the level of aortic trifurcation caudal to the emergence of the deep iliac circumflex artery. They were usually in a dorsal-lateral position with respect to the external iliac artery. Finally, at least two jejunal lymph nodes were visualized and measured for each subject and were usually found along the course of the artery or cranial mesenteric vein (Figure 7).

### 3.1. Lymph Node Size (Length, Thickness, Width) and Volume

Table 1 shows measurements regarding the length, thickness, width, and volume of the AL lymph nodes stratified according to body weight and localization.

Significant positive correlations between size of the lesion and BW were found for any kind of measurements in any localization. Strong relationships were found in almost all cases (ρ values less than 0.5 were found for thickness and width of gastric lesions only).

### 3.2. Lymph Node Shape

Some lymph nodes demonstrated a clear frequency of elongated shapes, such as jejunal lymph nodes and medial iliac (97.8% and 92.2% of cases, respectively), and hepatic lymph nodes, which showed a slightly lower frequency (66.3%). In gastric lymph nodes and in duodenal pancreatic lymph nodes, a rounded shape was more represented (frequency of 77.3% and 62.2%, respectively) than in the other nodes. The splenic lymph nodes showed a frequency for a mixed shape (43.2%) similar to that for an elongated shape (38.6%).

The distribution of BW according to the different shapes of the lymph nodes examined was described in Table 2. Significant, but weak, relationships between BW and shape were found in gastric and medial iliac lymph nodes only.

### 3.3. X-ray Attenuation and Enhancement

Observations concerning X-ray attenuation before and after contrast administration on the totality of ALs (*n* = 402) are reported in Table 3 according to AL body weight categories. No significant relationship between X-ray attenuation and continuous values of BW was found except a weak negative relationship in post-contrast attenuation of pancreatic duodenal nodes only (*p* = 0.047).

Regarding the distribution of the contrast medium, AL enhancement exhibited slightly higher percentages for homogeneous distribution (*n* = 273; 67.9%). BW was not significantly different between homogeneous and heterogeneous AL enhancement (Table 4).

### 3.4. Age

Finally, Table 5 shows the effect of age on ALs size and X-ray attenuation stratified according to body weight and localization. Significant correlation (Spearman rank correlation) was noted in splenic ALs volume (*p* = 0.007), pancreatic lymph nodes thickness, volume, and HU post contrast values (*p* = 0.035, *p* = 0.045, and *p* = 0.047, respectively), medial iliac lymph nodes length, width, and volume (*p* = 0.025, *p* = 0.029, and *p* = 0.018, respectively), as well as, jejunal lymph nodes length (*p* = 0.018), thickness, width, and volume (*p* < 0.001 each).

## 4. Discussion

One of the main objectives of this research is to study a possible relationship between AL CT measurements that are most visible and body weight in dogs without any sign of lymphadenopathy found through clinical evaluation, routine hematobiochemical analysis, and imaging. The population’s specimens included had good variability in breed and weight but showed a homogeneous distribution between groups S, M, and L regarding different variables, such as age (*p* = 0.36), sex or sterilization (*p =* 0.25, *p =* 0.31, respectively). In this ideal population, dogs with BCS lower than 2/5 and higher than 4/5 were excluded to avoid any differences regarding lymph nodes evaluation and visibility related to the abdominal fat [16]. Statistical analysis revealed a significant increase in AL measurements from a dog group with a lower weight to greater weight. Data obtained from all group comparisons showed statistical significance (*p* < 0.05) for each measurement performed, confirming the initial hypothesis. In anticipation of achieving this result, what is more interesting is which of these measures is more reliable to carry out a dimensional assessment. Although there is a good correlation between body weight and length, thickness, and width (*p* < 0.05) regarding with the volume, the significance were the most reliable (*p* < 0.001). Both for the authors’ experience and significance obtained the possibility of easily calculating the volume is an excellent means for a dimensional evaluation. Here, 3D volume rendering reconstructions made available by various software programs, in addition to expressing the numerical value of the volume, quickly revealed a shape to observers; this technology also allows evaluation outside other abdominal structures that may interfere with interpretation. From a single comparison between weight classes, another interesting analysis result was obtained. Although values were significantly increasing in multiple contracts between all groups, this did not always happen by comparing them to pairs. Hepatic lymph nodes, for example, showed only slight differences (*p* > 0.05) in length values between dogs weighing between 10 and 30 kg (M) and dogs weighing more than 30 kg (L). In splenic ones instead of width, significant data were obtained (*p* < 0.001) only for comparison between dogs of the S group and those of the L group. Indeed, subjects in the M group were coupled with those in the S group, and the L group revealed greater error possibilities (*p* > 0.05) than in the other groups. The same happened when comparing the length between categories M and L or the thickness between group S and group M (*p* > 0.05). The gastric lymph nodes showed insignificant data for the thickness between specimens weighing less than 10 kg (S) and those weighing between 10 and 30 kg (M), and no measurement was found to be reliable comparing dogs of group M and those of L. The same outcome also occurred for pancreaticoduodenal lymph nodes, except for the volume (*p* < 0.05). Finally, for the medial iliac and jejunal lymph nodes, significant values were maintained in all measurements (*p* < 0.05) in comparison between the two body weight categories. Certainly, volume size, even by latter estimate, is the most reliable value. Except for gastric lymph nodes, it was the only value that always showed little chance of error (*p* < 0.05).

Taking as population under consideration ALs regardless of location and weight divisions, the results obtained from attenuation and enhancement are very similar to those already present in the literature [16]. However, the goal of interest is to confirm that these characteristics are not actually affected by the body weight. In Group S, M, and L, and in those in which ALs were grouped by location, there were no values with significant differences (*p* > 0.05). Different Hounsfield values in these structures are random, and feedback for ALs is compatible with the classical attenuation values obtained by soft tissue. Regarding enhancement, there are no significant data. The expectation of the authors, for subjects that were presumably normal, was to obtain most of the investigated structures with homogeneous capturing, as described recently for sternal lymph nodes [14,18]. In most structures, the contrast medium was distributed homogeneously, but there were also a large number of lymph nodes with a heterogeneous distribution. Regarding this study, it must be remembered that a heterogeneous enhancement is also assigned to those structures in which one could easily observe a distinction between the cortical and medullary regions and therefore not necessarily identify a pathological process in progression expression. Heterogeneous, “ring” or absent enhancement of ALs in CT exams has been described in veterinary [11,13,19,20,21] and in human medicine [22,23,24,25,26], usually as an expression of pathological processes such as inflammation or metastasis. Care should be taken, however, because homogenous enhancement is not specific to normality, and heterogeneity is not exclusive to a pathological process. Evaluating only a homogeneous distribution is therefore insufficient; possible enhancement patterns are many, there are mixed patterns, the patterns may be peripheral, the lymph nodes may not acquire contrast to be subsequently normal or may be affected by inflammatory or tumor processes. It is therefore important to evaluate this factor always in relation to other features (for example, shape, and size). Additionally, shape evaluation shows few differences from what has already been treated [16]. This work, however, checks whether there is a correlation between the weight and AL length, which then results in an elongated shape. The values obtained were not significant for any of the investigated locations, except for the gastric lymph nodes (*p* < 0.05). The shapes of these structures are therefore in almost all cases an anatomical feature invariant under normal conditions, despite the body weight differences between the examined dogs.

The ultimate goal is to observe measurement behaviour obtained by separating younger from older dogs while maintaining weight differences. One of the inclusion criteria was reaching 18 months of age, so this study did not include puppies. Young subjects are animals that were suspected to be aged between 18 and 24 months. The same measurements described above were used to compare length, thickness, width, and volume between these two subgroups (youths vs. adults). The reason for this distinction is to ensure that these subjects have no age-related differences concerning the AL size; therefore, a dog aged 18 months can presumably be considered an adult in the context of CT-guided measurements of some ALs. Comparison does not raise any particular significance to many of the ALs involved. Regarding jejunal lymph nodes only, the data showed significant values in each weight categories and for different sizes. There seems to be a positive association regard to length (*p* < 0.05) thickness, width, and volume (*p* < 0.001). Therefore, considering volume to be the most reliable metric, it is reasonable to think that there is a positive correlation between subject’s young age (18–24 months) and a higher volumetric value only for jejunal lymph nodes. This possibility is also analyzed in human medicine; for example, an increase in the mesenteric lymph node size found with CT in pediatric subjects (range 1.1–17.3 years) is considered a non-specific lymphadenopathy finding. Furthermore, for children, a maximum dimensional limit slightly higher than adult subjects is set [27].

The main limitation of the present study was related to its retrospective nature. A comparison between tomographic and ultrasonographic measurements may provide further details.

## 5. Conclusions

In conclusion, we have verified with this research that dogs exhibit some significant changes in the normal abdominal lymph node size according to the body weight. These data can be used in dogs as a reference for size and appearance of hepatic, splenic, gastric, duodenal pancreatic, medial, and jejunal iliac lymph nodes in relation to their body weight, and thus lay foundations for a comparison with the results obtained from unhealthy subjects. Through these evaluations, we can therefore differentiate with certainty subjects suffering from inflammatory or tumor processes in the abdomen or in other regions. Our proposal is therefore to insert new normal dimensional parameters, catalogued according to the dog size, regarding the length, width, thickness, volume, and shape in order to evaluate them in future studies and routine clinical investigations in consideration of a variable such as body weight whose importance cannot be underestimated in veterinary medicine.

## Figures and Tables

**Figure 1 vetsci-08-00044-f001:**
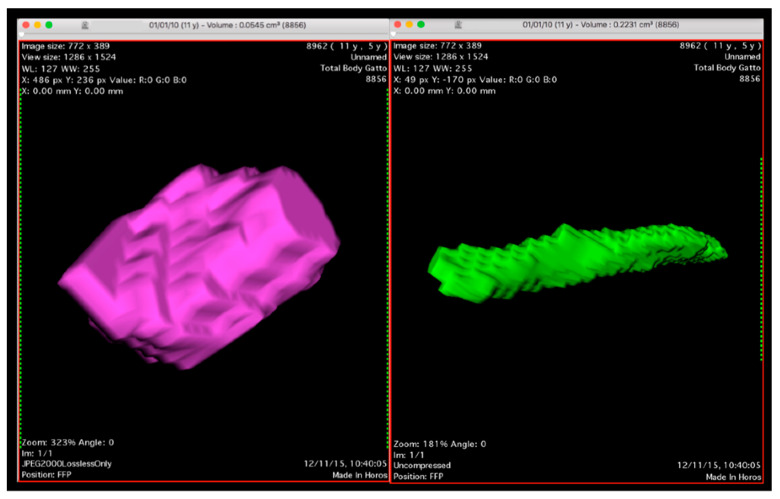
3D Volume Rendering sample of an elongated jejunal (**right**) and rounded gastric lymph node (**left**) calculated by software according to the procedure described.

**Figure 2 vetsci-08-00044-f002:**
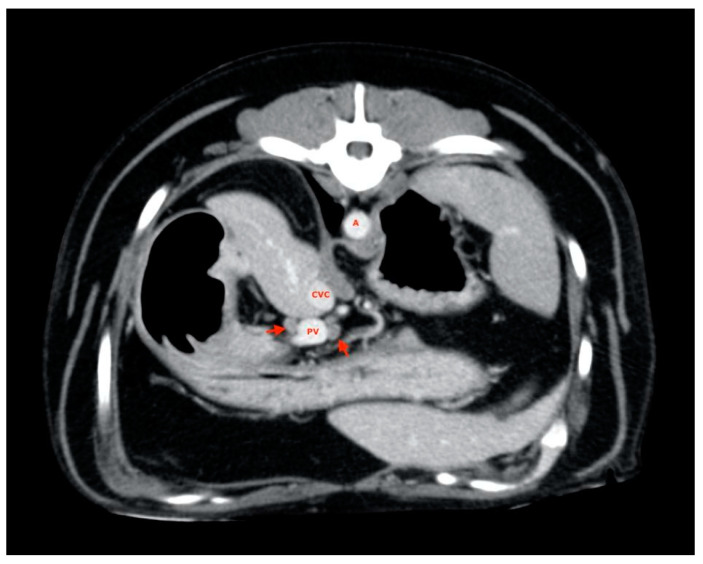
Hepatic lymph nodes (red arrow). A, aorta; CVC, caudal vena cava; PV, portal vein.

**Figure 3 vetsci-08-00044-f003:**
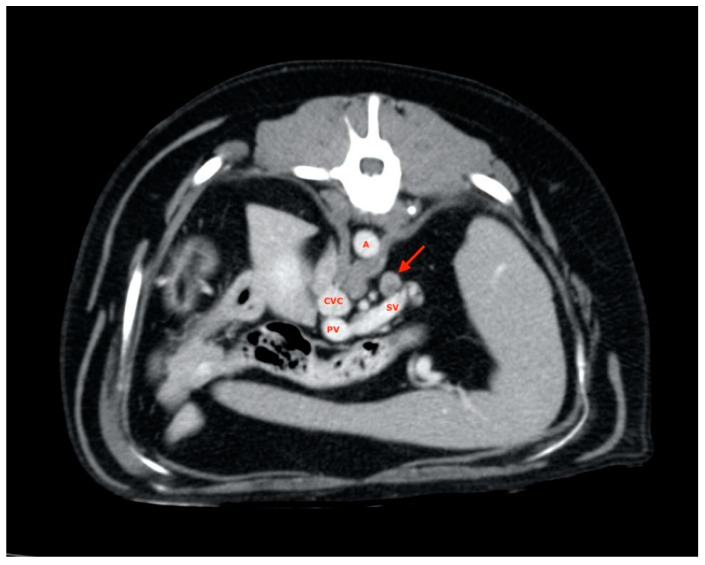
Splenic lymph node (red arrow). A, aorta; CVC, caudal vena cava; PV, portal vein; SV, splenic vein.

**Figure 4 vetsci-08-00044-f004:**
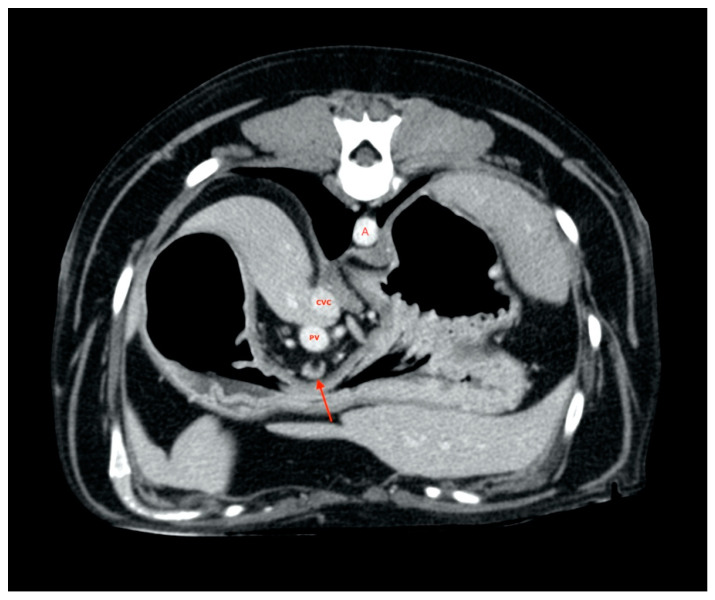
Gastric lymph node (red arrow). A, aorta. CVC, caudal vena cava. PV, portal vein

**Figure 5 vetsci-08-00044-f005:**
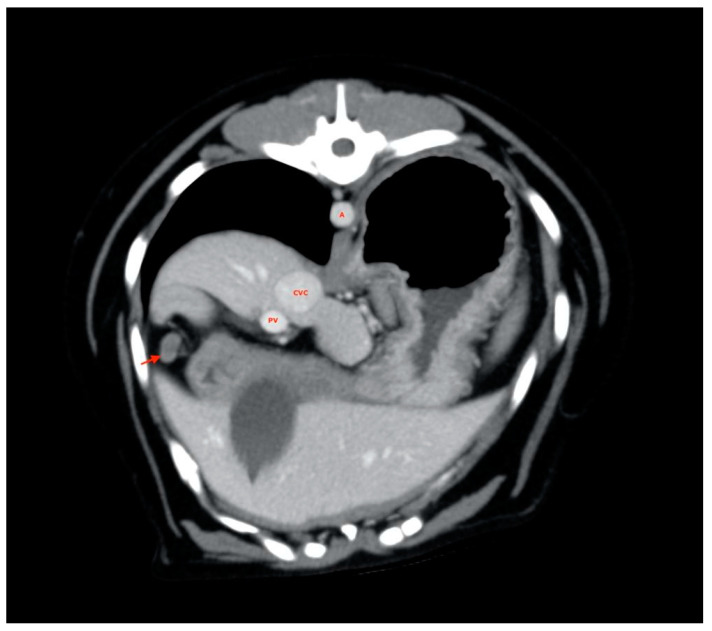
Pancreaticoduodenal lymph node (red arrow). A, aorta; CVC, caudal vena cava; PV, portal vein.

**Figure 6 vetsci-08-00044-f006:**
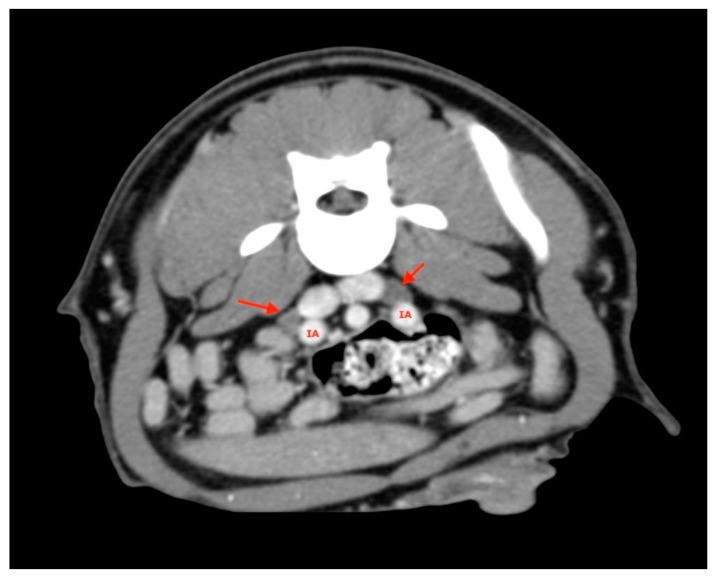
Medial Iliac lymph nodes (red arrows). IA, iliac artery.

**Figure 7 vetsci-08-00044-f007:**
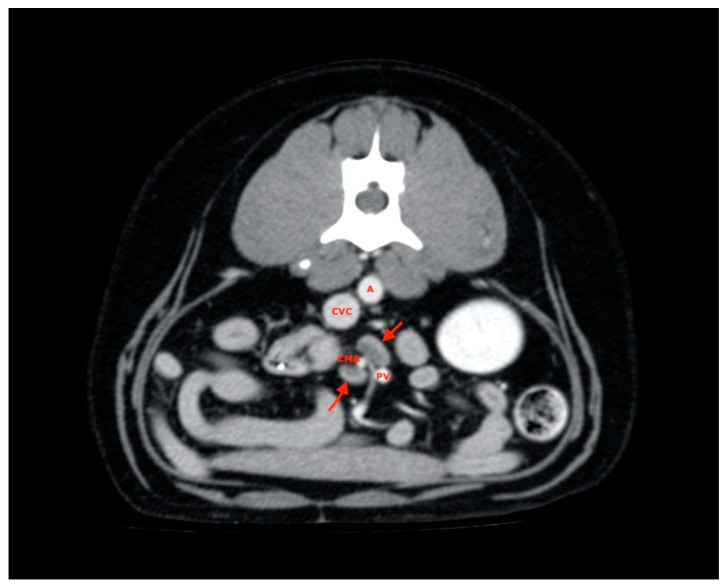
Jejunal lymph nodes (red arrows). A, aorta; CVC, caudal vena cava; PV, portal vein; CMA, cranial mesenteric artery.

**Table 1 vetsci-08-00044-t001:** Median and range (within parentheses) values of ALs size according to body weight categories. Data are stratified by localization.

Lymph Nodes	Length (mm)	Thickness (mm)	Width (mm)	Volume (mm^3^)
**Hepatic:** *Total (n = 89)*	19.4 (4.5–59.7) **ρ = 0.552; *p* < 0.001**	4.9 (1.9–11.2) **ρ= 0.712; *p* < 0.001**	5.3 (2.0–10.7) **ρ = 0.668; *p* < 0.001**	313 (12–1753) **ρ = 0.635; *p* < 0.001**
Small: <10 kg (*n* = 30)	13.8 (4.5–33.0)	3.5 (1.9–5.6)	3.5 (2.0–6.2)	146 (12–728)
Medium: 10–30 kg (*n* = 30)	21.0 (5.8–56.8)	4.8 (2.5–8.2)	5.5 (2.8–8.3)	360 (28–1.028)
Large: >30 kg (*n* = 29)	27.3(7.3–59.7)	6.2 (4.4–11.2)	6.8 (3.7–10.7)	520 (105–1753)
**Splenic:** *Total (n = 44)*	14.7 (5.1–25.1) **ρ = 0.509; *p* < 0.001**	4.5 (2.1–13.7) **ρ = 0.668; *p* < 0.001**	5.1 (2.7–15.6) **ρ = 0.630; *p* < 0.001**	200 (27–967) **ρ = 0.833; *p* < 0.001**
Small: <10 kg (*n* = 15)	9.5 (5.1–17.0)	3.9 (2.1–5.6)	4.1 (2.9–7.2)	67 (27–294)
Medium: 10–30 kg (*n* = 15)	15.3 (8.8–25.1)	5.0 (3.3–10.4)	5.7 (2.2–10.2)	214 (111–482)
Large: >30 kg (*n* = 14)	18.5 (11.1–24.2)	7.0 (4.5–13.7)	6.9 (4.5–15.6)	435 (127–967)
**Gastric:** *Total (n = 44)*	9.6 (2.2–19.7) **ρ = 0.737; *p* < 0.001**	5.1 (2.1–13.1) **ρ = 0.407; *p* = 0.006**	5.7 (2.1–11.2) **ρ = 0.427; *p* = 0.004**	157 (11–490) **ρ = 0.660; *p* < 0.001**
Small: <10 kg (*n* = 14)	5.5 (2.2–9.9)	3.8 (2.1–7.2)	4.8 (2.1–6.8)	36 (11–194)
Medium: 10–30 kg (*n* = 15)	10.8 (5.4–16.0)	5.3 (3.8–11.5)	5.3 (3.8–11.2)	170 (54–450)
Large: >30 kg (*n* = 15)	12.1 (5.9–19.7)	6.5 (3.0–13.1)	6.5 (3.7–11.0)	301 (46–490)
**Pancreaticoduodenal:** *Tot* *al (n = 45)*	9.1 (2.4–24.2) **ρ = 0.572; *p* < 0.001**	5.8 (2.6–10.6) **ρ = 0.603; *p* < 0.001**	4.4 (1.9–12.6) **ρ = 0.565; *p* < 0.001**	109 (12–580) **ρ = 0.756; *p* < 0.001**
Small: <10 kg (*n* = 15)	7.1 (2.4–10.1)	3.8 (2.6–7.8)	3.5 (1.9–6.2)	40 (12–164)
Medium: 10–30 kg (*n* = 15)	9.2 (4.4–20.0)	6.0 (4.3–9.7)	4.4 (3.2–9.5)	214 (60–319)
Large: >30 kg (*n* = 15)	13.0 (5.8–24.2)	7.1 (3.2–10.6)	6.6 (3.0–12.6)	435 (74–580)
**Medial iliac:** *Total (n = 90)*	24.6 (6–57) **ρ = 0.762; *p* < 0.001**	5.2 (1.9–13.9) **ρ = 0.750; *p* < 0.001**	4.9 (1.5–11.2) **ρ = 0.622; *p* < 0.001**	354 (32–1987) **ρ = 0.802; *p* < 0.001**
Small: <10 kg (*n* = 30)	16.5 (6.0–56.5)	3.8 (1.9–6.9)	4.1 (1.5–8.5)	135 (32–713)
Medium: 10–30 kg (*n* = 30)	24.5 (17.7–55.9)	5.0 (2.5–11.1)	5.0 (2.5–11.1)	214 (124–1272)
Large: >30 kg (*n* = 30)	33.1 (20.0–57.0)	7.2 (4.0–14.0)	6.8 (3.4–11.2)	844 (233–1987)
**Jejunal:** *Total (n = 90)*	50 (7.7–89.3) **ρ = 0.748; *p* < 0.001**	5.8 (2.4–11.6) **ρ = 0.551; *p* < 0.001**	6.5 (2.3–15.8) **ρ = 0.640; *p* < 0.001**	1498 (85–5273) **ρ = 0.817; *p* < 0.001**
Small: <10 kg (*n* = 30)	34.2 (7.7–58.1)	4.0 (2.4–8.0)	4.2 (2.3–8.1)	451 (85–1532)
Medium: 10–30 kg (*n* = 30)	53.0 (28.0–79.0)	5.9 (4.1–8.7)	7.0 (2.4–11.8)	959 (287–3965)
Large: >30 kg (*n* = 30)	65.0 (40.9–89.3)	7.1 (3.3–11.6)	9.2 (4.8–15.8)	2.6 (1.03–5.3)

The Spearman rank correlation test was used in order to test the relationships between ALs and dog size. Significant ρ and *p* values are highlighted by using bold characters.

**Table 2 vetsci-08-00044-t002:** Median and range (within parentheses) values of BW (kg) according to different shapes in the lymph nodes examined. Data were stratified by localization.

Lymph Nodes	Rounded ALs	Mixed ALs	Elongated ALs	Spearman Rank Test
Hepatic (*n* = 89)	30.5 (4.5–38)	20.5 (2–62)	18 (2–62)	ρ = −0.051; *p* = 0.634
Splenic (*n* = 44)	31.5 (9–38)	20 (2–62)	14 (5–50)	ρ = −0.230; *p* = 0.134
Gastric (*n* = 44)	15 (2–44)	30 (14–62)	-	**ρ = 0.380; *p* = 0.011**
Pancreaticoduodenal (*n* = 45)	15 (2.5–62)	22 (2–44)	25 (23.5–32)	ρ = 0.049; *p* = 0.751
Medial iliac (*n* = 90)	-	4.5 (2–42)	23.5 (2.5–62)	**ρ = 0.221; *p* = 0.037**
Jejunal (*n* = 90)	-	6 (6–6)	21.7 (2–62)	ρ = 0.192; *p* = 0.070

The Spearman rank correlation test was used in order to test the relationships between the progression from rounded to elongated lymph node shape and dog size. Significant ρ and *p* values are highlighted by using bold characters.

**Table 3 vetsci-08-00044-t003:** Median and range (within parentheses) values of X-ray attenuation of ALs before and after contrast administration according to body weight categories. Data are stratified by localization.

Lymph Node	HU Pre MdC Median (Range)	HU Post MdC Median (Range)
**Hepatic:** *Total (n = 89)* Small: <10 kg (*n* = 30) Med: 10–30 kg (*n* = 30) Large: >30 kg (*n* = 29)	29 (20–49) ρ = 0.130; *p* = 0.224 26 (20–40) 31.5 (20–49) 31 (20–49)	94 (73–113) ρ = 0.117; *p* = 0.274 26 (20–40) 31.5 (20–49) 31 (20–49)
**Splenic:** *Total (n = 44)* Small: <10 kg (*n* = 15) Med: 10–30 kg (*n* = 15) Large: >30 kg (*n* = 14	29.5 (19–43) ρ = 0.053; *p* = 0.732 27 (19–33) 32 (21–43) 28.5 (19–37)	91.5 (70–129) ρ= −0.011; *p* = 0.942 96 (75–129) 91 (76–104) 90.5 (70–118)
**Gastric:** *Total (n = 44)* Small: <10 kg (*n* = 14) Med: 10–30 kg (*n* = 15) Large: >30 kg (*n* = 15)	24 (19–38) ρ = 0.192; *p* = 0.212 22 (19–37) 25 (21–31) 27 (21–38)	89.5 (73–114) ρ= −0.027; *p* = 0.860 88 (77–114) 14 (75–101) 89 (73–102)
**Pancreaticoduodenal:** *Total (n = 45)* Small: <10 kg (*n* = 15) Med: 10–30 kg (*n* = 15) Large: >30 kg (*n* = 15)	27 (19–38) ρ= −0.012; *p* = 0.938 28 (20–35) 27 (21–38) 27 (19–37)	90 (71–129) **ρ= −298; *p* = 0.047** 97 (79–121) 91 (78–103) 87 (71–129)
**Medial iliac:** *Total (n = 90)* Small: <10 kg (*n* = 30) Med: 10–30 kg (*n* = 30) Large: >30 kg (*n* = 30)	27 (19–43) ρ = 0.052; *p* = 0.629 27 (20–32) 29 (19–41) 26.5 (19–43)	89.5 (70–147) ρ = 0.033; *p* = 0.755 92.5 (70–108) 89 (71–147) 93 (33–132)
**Jejunal:** *Total (n = 90)* Small: <10 kg (*n* = 30) Med: 10–30 kg (*n* = 30) Large: >30 kg (*n* = 30)	28 (19–48) ρ = 0.180; *p* = 0.090 26 (20–32) 32 (21–48) 28.5 (19–42)	98.0 (70–156) ρ= −0.036; *p* = 0.735 96 (71–120) 102 (70–156) 95.5 (77–121)

The Spearman rank correlation test was used in order to test the relationships between X-ray attenuation of ALs and dog size. Significant **ρ** and *p* values are highlighted by using bold characters.

**Table 4 vetsci-08-00044-t004:** Median and range values of BW (kg) according to ALs enhancement stratified according to localization.

Lymph Nodes	Enhancement		Kruskal–Wallis Test
	Homogeneus	Heterogeneous	
Hepatic (*n* = 89)	23.5 (2–62)	15 (2.5–50)	Χ^2^ = 0.019; *p* = 0.889
Splenic (*n* = 44)	19 (2.5–44)	18.5 (2–62)	Χ^2^ = 0.036; *p* = 0.850
Gastric (*n* = 44)	21.7 (2–50)	22 (2.5–62)	Χ^2^ = 0.020; *p* = 0.889
Pancreaticoduodenal (*n* = 45)	21.7 (2–44)	9 (2.5–62)	Χ^2^ = 0.057; *p* = 0.812
Medial iliac (*n* = 90)	20 (2–50)	30 (2.5–62)	Χ^2^ = 0.099; *p* = 0.753
Jejunal (*n* = 90)	23.5 (2–62)	16 (2.5–42)	Χ^2^ = 0.017; *p* = 0.897

The Kruskal–Wallis test was used in order to test the difference of dog size between homogeneous and heterogeneous enhancement. No significant *p* values were found.

**Table 5 vetsci-08-00044-t005:** Median and range (within parentheses) values of ALs size and attenuation according to age and body weight categories. Data are stratified by localization.

	Age (No.)	Length (mm)	Thickness (mm)	Width (mm)	Volume (mm^3^)	HU Pre MdC	HU Post MdC
**Hepatic**		ρ = −0.089 *p* = 0.407	ρ = −0.140 *p* = 0.192	**ρ = −0.279** ***p* = 0.008**	ρ = −0.086 *p* = 0.422	ρ = −0.013 *p* = 0.904	ρ = −0.120 *p* = 0.264
Small: <10 kg (*n* = 30)	Y (*n* = 6) A (*n* = 24)	14.7 (6.7–23.5) 13.3 (4.5–33.0)	3.3 (2.1–5.6) 3.5 (1.9–5.4)	3.1 (2.5–6.2) 3.6 (2–6)	137.9 (25.4–457.2) 146.4 (11.7–728.8)	27.5 (21–35) 26 (20–40)	93.5 (90–100) 93 73–108)
Medium:10–30 kg (*n* = 30)	Y (*n* = 6) A (*n* = 24)	23 (12.2–27.8) 20.2 (5.8–56.8)	5.5 (2.7–6.4) 4.7 (2.5–4.2)	6.1 (4.6–69) 5.4 (2.8–8.3)	366.3 (161.3–561.2) 341 (28.5–1028)	30 (21–49) 32 (20–46)	100.5 (78–113) 94 (77–111)
Large: >30 kg (*n* = 29)	Y (*n* = 11) A (*n* = 18)	32.2 (7.3–59.7) 25.9 (13.1–51.7)	7.7 (4.4–11) 5.9 (4.6–11.2)	7.4 (5–7.7) 6.4 (3.7–9.2)	386 (105.4–1754) 524.5 (151–1073)	31 (20–49) 30 (21–41)	97 (77–108) 96.5 (78–112)
**Splenic**		ρ = −0.229 *p* = 0.134	ρ = −0.174 *p* = 0.258	ρ = −0.171 *p* = 0.266	**ρ = −0.399** ***p* = 0.007**	ρ = 0.170 *p* = 0.270	ρ = −0.134 *p* = 0.386
Small: <10 kg (*n* = 15)	Y (*n* = 3) A (*n* = 12)	9.8 (5.6–16.9) 8.8 (5–16.7)	3.5 (2.7–5.6) 4 (2–4.5)	4.3 (3.7–4.6) 4 (2.9–7.2)	111 (27.7–294) 64.4 (28.2–140.6)	24 (23–30) 28 (19–33)	101 (96–103) 92 (75–129)
Medium:10–30 kg (*n* = 15)	Y (*n* = 3) A (*n* = 12)	18.4 (11.9–25.1) 14.9 (88–24.6)	5.8 (3.8–5.9) 4.4 (3.3–10.4)	6.7 (4–7.4) 5.4 (2.7–10.2)	266.5 (250.9–299.6) 206.8 (111.2–482.3)	22 (21–34) 32 (23–43)	91 (88–94) 89 (56–104)
Large: >30 kg (*n* = 14)	Y (*n* = 6) A (*n* = 8)	18.5 (11.2–18.4) 17 (11–24.3)	7.6 (4.5–13.7) 7 (4.6–11.2)	5.8 (4.5–13.6) 7.2 (4.5–10)	483.3 (167.9–651.2) 389.7 (127.6–967)	25 (23–37) 32 (19–37)	89.5 (70–106) 93.5 (78–118)
**Gastric**		ρ = −0.242 *p* = 0.113	ρ = −0.275 *p* = 0.071	ρ = −0.143 *p* = 0.356	ρ = −0.268 *p* = 0.079	ρ = −0.091 *p* = 0.557	ρ = −0.249 *p* = 0.103
Small: <10 kg (*n* = 14)	Y (*n* = 3) A (*n* = 11)	5.5 (4.2–6.2) 5.5 (2.2–9.9)	6.7 (4.2–6.8) 3.5 (2–6.8)	5.6 (3.5–7.2) 3.4 (2–6.9)	116.7 (27.8–194.7) 36.4 (11–122.9)	24 (23–37) 22 (11–27)	92 (81–114) 88 (77–97)
Medium:10–30 kg (*n* = 15)	Y (*n* = 3) A (*n* = 12)	12.3 (7–12.5) 10 (5.4–16)	4.9 (3.9–8.4) 5.5 (3.8–11.2)	5.8 (5–5.8) 5.2 (3.8–5.5)	196 (54.9–210.6) 169 (74.3–449.7)	24 (21–27) 25.5 (22–31)	86 (86–89) 86.5 (75–101)
Large: >30 kg (*n* = 15)	Y (*n* = 6) A (*n* = 9)	13 (11.2–19.7) 12 (5.9–16.3)	7.1 (4.9–10.9) 6.6 (3.7–9.8)	6 (3.9–11.8) 7.1 (3–13.1)	285.4 (124.7–398.2) 301.7 (46.7–489.7)	24 (22–28) 29 (21–38)	95.5 (89–102) 83 (73–98)
**Pancreatic duodenal**		ρ = −0262 *p* = 0.083	**ρ = −0.314** ***p* = 0.035**	ρ = −0.183 *p* = 0.228	**ρ = −0.301** ***p* = 0.045**	ρ = −0.022 *p* = 0.885	**ρ = 0.297** ***p* = 0.047**
Small: <10 kg (*n* = 15)	Y (*n* = 3) A (*n* = 12)	8 (5.7–9.2) 7 (2.4–10.1)	4 (2.6–7.8) 3.8 (2.8–5.3)	5.1 (2.9–6.2) 3.4 (1.9–5.4)	108.2 (29.3–164.9) 55.1 (12.7–100.8)	28 (22–35) 27 (20–33)	90 (88–92) 98 (79–101)
Medium:10–30 kg (*n* = 15)	Y (*n* = 3) A (*n* = 12)	9.3 (8.4–9.8) 9.15 (4.4–20)	9.4 (6.7–9.7) 5.8 (4.3–8.4)	5.8 (4.4–6.4) 4.2 (3.2–9.5)	189.8 (91.3–319.3) 158.8 (60.5–309.9)	25 (23–29) 29 (21–38)	89 (78–91) 94.5 (79–103)
Large: >30 kg (*n* = 15)	Y (*n* = 6) A (*n* = 9)	13 (5.8–19.9) 13 (6.2–24.1)	7.2 (5.9–9.1) 6.6 (3.2–10.6)	4.9 (3–9.1) 6.7 (4–12.6)	318.5 (74.6–580.8) 266.1 (98.8–399.3)	29.5 (19–33) 25 (19–37)	92.5 (78–101) 86 (71–129)
**Medial iliac**		**ρ = −0.236** ***p* = 0.025**	ρ = −0.149 *p* = 0.161	**ρ = −0.230** ***p* = 0.029**	**ρ = −0.249** ***p* = 0.018**	ρ = 0.155; *p* = 0.145	ρ = 0.147 *p* = 0.166
Small: <10 kg (*n* = 30)	Y (*n* = 6) A (*n* = 24)	16.6 (6–25.4) 16.6 (6.5–56.5)	3.6 (2.9–4.3) 3.8 (1.9–6.9)	4.4 (4.1–5.9) 3.6 (1.5–6.5)	146.2 (52–404.7) 126.7 (32.2–713.1)	26 (22–32) 27 (20–32)	92 (70–106) 92.5 (76–108)
Medium:10–30 kg (*n* = 30)	Y (*n* = 6) A (*n* = 24)	24.2 (20–55.9) 24.5 (17.7–37.5)	5.2 (2.6–8.4) 5.4 (3.4–7.7)	4.9 (2.5–8.2) 5.2 (3.1–11.1)	292 (191.8–1273) 346.2 (124.4–867.1)	25.5 (19–37) 30 (21–41)	88 (78–138) 89.5 (71–147)
Large: >30 kg (*n* = 30)	Y (*n* = 12) A (*n* = 18)	32.4 (24.9–52.3) 33.4 (20.1–57.2)	7.7 (4.3–10.2) 7 (4–13.9)	7.3 (6.1–9.6) 5.8 (3.4–11.2)	1098 (407–1988) 685.1 (233.5–1875)	24 (21–33) 29 (19–43)	85 (33–108) 96 (77–132)
**Jejunal**		**ρ = −0.248** ***p* = 0.018**	**ρ = −0.405** ***p* < 0.001**	**ρ = −0.420** ***p* < 0.001**	**ρ = −0.478** ***p* < 0.001**	ρ = 0.004 *p* = 0.971	ρ = 0.182 *p* = 0.087
Small: <10 kg (*n* = 30)	Y (*n* = 6) A (*n* = 24)	33.6 (25.1–43.8) 34.7 (7.7–58.1)	5.3 (3.6–7.2) 3.9 (2.4–8)	5.8 (3.2–8) 4.1 (2.3–8.1)	528.3 (274.2–1532) 349.5 (85.7–873.5)	28 (21–32) 25 (20–29)	102.5 (94–109) 95 (71–120)
Medium:10–30 kg (*n* = 30)	Y (*n* = 6) A (*n* = 24)	60.5 (45.3–70.3) 50.6 (28–78.9)	6.9 (5.6–8.7) 5.7 (4.1–8.7)	9.6 (7.5–11.8) 6.2 (2.4–9.4)	2215 (1179–3965) 1499 (287.5–3739)	26 (21–34) 32 (21–48)	83 (70–102) 108 (70–156)
Large: >30 kg (*n* = 30)	Y (*n* = 12) A (*n* = 18)	71.1 (42.2–84) 63.8 (40.9–89.3)	8.1 (6.1–11.6) 6.4 (3.3–8.9)	9.7 (6.6–15.8) 8 (4.8–12.4)	3634 (2613–5273) 2075 (1029–4380)	28.5 (21–35) 28.5 (19–42)	95.5 (77–121) 95 (78–111)

Y: Youths (18–24 months); A: Adults (>2 years); the Spearman rank correlation test was used in order to test the relationships between ALs size and attenuation and age. Significant **ρ** and *p* values are highlighted by using bold characters.

## Data Availability

The data presented in this study are available in this article.

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
