# Peer review of "Computed Tomography Evaluation of Normal Canine Abdominal Lymph Nodes: Retrospective Study of Size and Morphology According to Body Weight and Age in 45 Dogs"

_vetsci, 2021, doi:10.3390/vetsci8030044_

Round 1
Reviewer 1 Report
Dear authors,
I would like to thank you for your great efforts in improving the manuscript. Although improved, in my opinion, the manuscript is still presenting some statistical analysis flaws.
Please, see the attached peer-review file for more details.
Sincerely

Reviewer 2 Report
This manuscript was revised and improved. There is no more comments to the manuscript.
Reviewer 3 Report
Dear Authors,
I appreciate your changes following my suggestions.
I reread the entire modified paper, the topic is very interesting and its discussion is expressed with scientific rigor.
Kind regards
Reviewer 4 Report
Dear Authors,
your comments and changes improved significanlty the manuscript. I consider the manuscript accept in the present form.
This manuscript is a resubmission of an earlier submission. The following is a list of the peer review reports and author responses from that submission.
Round 1
Reviewer 1 Report
The authors evaluated the size and contrast enhancement of abdominal lymph nodes during CT examination in dogs presumably free of abdominal disease. Moreover, the authors evaluated the relationship between age, weight and lymph node features.
Although the study is interesting and may fill some literature gaps, the study design and statistical analysis put some uncertainty in the study results and conclusions.
See below for more details.
Line 47-53: I suggest to rephrase the sentence. "This study includes several adult dogs that did not show clinical signs attributable to lymphadenopathy and considers the examination of normal lymph nodes", Do the authors refer to the reference [11] or their own study? "as in a previous review", "precise inclusion criteria", "In addition to these features", please be more specific.
Line 53-56 "Considering limited information in the literature, the aim of this retrospective study was to establish the normal AL size in presumably healthy dogs according to body weight and age so that this size can provide a reference for evaluation of infectious or neoplastic events." The study did not seem to be designed to answer to the authors' hypothesis. This study evaluated the relationships between age, body weight, and lymph node features. I suggest to rephrase.
Line 62-75 The study population taken into account by the authors, put some uncertainty to the possibility to make inferences about other presumably healthy dogs. I would precise what the authors meant with the term presumably. Would they include sick dogs? How would they exclude sick dogs? Why did the authors only include dogs with a BCS ranged from 2 to 4? I think the authors may discuss the reasons of their inclusion criteria, and how dogs with other features (i.e BCS of 1 or 5) could have affected their results.
Line 76-82 The authors divided dogs into 3 categories according to the body weight. Why don't they analyse BW as a continuous variable? Correlation tests may reduce number of test (and type I error) and may reduce the error due to the empirical stratification. The same is for age class divisions. I suggest to firstly run correlation tests, and then, re-evaluate stratification based on data analysis and distribution.
Line 104-105 How many operators measured the 3 dimensions of LN?
Line 106-109 "The length was defined as lymph node’s maximum size, regardless of position. The thickness was calculated by measuring the extension in a dorsoventral direction perpendicular to the length. The width was calculated by measuring extension in a laterolateral sense always perpendicular to the length." As I understand, the thickness was measured in a DV direction, the width in a LL direction so I guess the length was measured in a craniocaudal sense and not regardless of position as the authors stated. Please clarify.
Line 124-129 I would suggest to use only median and range instead of mean and SD, since all the statistical analysis performed in the present study compared medians and not means. There is no information regarding post-hoc tests or how the authors compared pairs of groups. Moreover, the present study collected reapeted measures but data were analyzed as they were indipendent. I suggest to the authors to consult a statistician to confirm the goodness of their statistical analysis.
Line 131 I would change the term "apparently healthy". A dog with a bone fracture does not sound "healthy".
Line 136-137 I suggest to add more information regarding reason and type of surgery of the 7 dogs examined for follow-up purpose. The reader should know how the previous disease can affect the results.
Line 138-160 I suggest to remove means and SD, leaving medians and ranges. I would remove definition of groups in brackets, the authors have already described the group features in the materials and methods section.
Figure 1 to Figure 6. I would thank the authors for their work. I find their figures and explanations very satisfying.
Table 1. This table is very hard to read. First, I would remove means and SDs. And I would remove exact p-values and use more symbols or letters. However, I would re-evaluate the table after revision of the statistical analysis.
Table 2. The authors stated they used the Kruskal-Wallis test, however this table shows contingency tables. Here it is clearer what I meant when I talked about reapeted measures. The authors included 45 dogs, however they seemed to evaluate 89 dogs when the association between hepatic LN shape and BW category was investigated. However, I would re-evaluate the table after revision of the statistical analysis.
Line 218 Means lack of unit of measure
Table 3. Again, pre and post contrast values are reapeted measures and should be analyzed with other tests. I would re-evaluate the table after revision of the statistical analysis.
Table 4. The authors stated they used the Kruskal-Wallis test, however this table shows contingency tables. I would re-evaluate the table after revision of the statistical analysis.
Table 5. This table is the hardest table to be read. Many p-values, however no information regarding which type of comparison/association was tested has been provided. I would re-evaluate the table after revision of the statistical analysis.
Discussion section. Since results would change after the statistical analysis revision, I can not provide any specific comments to the authors. I suggest to add a few lines regarding limitations of the study at the end of the discussion section. Moreover I suggest the authors to reduce the number of tests (Type I error), and that more categories mean larger contingency tables with smaller number in each cell that can result in Type II errors.
Author Response
Dear Reviewer,
thank you very much for your comments and suggestions.
Please find attached our rebuttal.
Kind regards

Reviewer 2 Report
There are some questions about this study as follows.
1) Page 3, Line 128-129: Please specify the exact version of the program used in the statistics of this study (SPSS version)
2) Page 3, Line 135-137:
The indications for CT were 34 dogs with acute thoraco-lumbar spine injury as discopathies or traumatic events (subluxation, luxation or fracture); 4 dogs with Horner's syndrome were negative. In addition, 7 dogs were examined for follow-up six months after surgery.
In this study, authors described normal CT evaluation results of abdominal lymph nodes in dogs.
The key to this study is to show the results in normal.
But 34 are patients had severe spinal cord damage with trauma (fracture etc...).
Why did you put the case in study where there was trauma to the extent that severe damage to the spinal cord?
What surgery did the seven dogs undergo six months ago?
What means that 4 dogs with Horner's syndrome were negative?
It is a really good research paper. But if the selection of dogs used in this study is wrong, would not it be difficult to trust the results of this study?
Studies on the size of abdominal lymph nodes of normal dogs have already been quantified by ultrasound.
In dogs used in this experiment, the size of abdominal lymph nodes would have been better compared to ultrasound.
Author Response

(The authors gave the same response as above.)

Reviewer 3 Report
Dear Authors
I reviewed the manuscript entitled "Computed tomography evaluation of normal canine abdominal lymph nodes: retrospective study of size and morphology according to body weight and age in 45 dogs". The study is well designed and conveys informations about the size and shape of normal limph nodes in dogs, related to age and body weight. In my opinion the topic is interesting and can be useful to the reader. I have only few comments (see specific comments below), therefore I recommend minor revision
Specific comments:
line 99: please specify the post-contrast scan protocol. Looking at the figures, I assume that only the venous phase was considered.Please clarify.
line 101: Authors stated that: "The ALs analysed were two jejunal, one hepatic, one splenic, one gastric, one pancreaticoduodenal and two medial iliacs...". Please clarify why only one hepatic limph node was considered.
line 110:"The area was calculated on transverse images by the perimeter of a selected structure. This calculation was performed on every single consecutive image that included the selected structure. The perimeter was defined manually"... It is unclear to me how the volume was calculated. Please clarify. Adding a figure could be helpful.
line 113: "Shape was judged in a subjective way through 3D volume rendering reconstructions with an evaluation similar to that reported for ultrasonography" : please add a figure in order to clarify this sentence.
line 259: "Here, 3D volume rendering reconstructions made available by various software programs, in addition to expressing the numerical value of the volume, quickly revealed a shape to observers". It is unclear the role of 3D volume rendering in calculating the volume (see comments above).
line 257: ...In splenic tissue instead of width...Did you mean splenic nodes?
Author Response

(The authors gave the same response as above.)

Reviewer 4 Report
Thank you for your contribution to the advancement of veterinary medicine with your research, as you described in the article 'Computed tomography evaluation of normal canine abdominal lymph nodes: retrospective study of size and morphology according to body weight and age in 45 dogs'.
I would kindly ask for two clarifications.
Materials and Methods
- lines 86 and 87: it is not specified if the maintenance with isofluorane was mixed with oxygen.
- line 96: I'd like to know if the sternal decubitus has been used for all the dogs, included the subjects with spinal injury.
I asked it because the line 135 indicates 34 dogs with this type of damage.
Author Response

(The authors gave the same response as above.)

Reviewer 5 Report
Dear Author,
your scientific paper is very interesting and the results obtained are useful in clinical practice. Minor revisions added in the attached pdf file are required. Revision of English is required

Author Response

(The authors gave the same response as above.)
